# Corporate Sustainability and Market Response According to the Name Change Strategy: Focusing on Korean IT Industry Firms

Jungmi Park and Yoojin Shin *

Division of Business Administration, Wonkwang University, Iksan 54538, Korea
* Correspondence: yooji01@wku.ac.kr; Tel.: +82-63-850-6288

**Abstract:** This study seeks to determine whether corporate sustainability increases due to the corporate name change strategy by analyzing the sample of IT companies listed on the KOSDAQ market from 2010 to 2019, through the event study methodology and OLS regression. This study has the following conclusions: first, the analysis results show that if the market response to an IT company's corporate name change is positive, its financial constraint improves after the name change. Second, even if the companies' financial constraint conditions before the corporate name change differ, their financial constraints improve after the name change if the market response to their announcements to change their corporate names is favorable. In other words, in terms of corporate sustainability, the financial constraint of a company improves depending on how it establishes the strategy to change its corporate name as well as the level of the market response to the announcement to change its corporate name. This implies that an IT company's strategy to change its corporate name can affect corporate sustainability. Consequently, from the perspective of IT companies, this study serves as a guide for stakeholders' decision-making processes and proves that the financial constraint can be improved through a corporate name change strategy.

**Keywords:** corporate name change; financial constraint; market reaction; IT companies

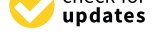



## 1. Introduction

The UK company, Interbrand, which is the world's leading brand consultancy, announces the "Best Global Brands" annually. Apple once again took the top spot in 2021, followed by Amazon, Microsoft, Google, and Samsung for the second to fifth spots, respectively. The total value of the selected top 100 brands in 2021 amounts to 2,667,524 million USD, which is a 15% increase from the previous year, and the fastest growing and most valued brands are those related to technology (https://interbrand.com/best-brands, accessed on 3 July 2022). In particular, all the top five brands are affiliated to the IT industry, and their brands are consistent with their corporate names. The frontrunner Apple's brand value is 408,251 million USD, and the brand value of Samsung, ranked fifth, is 74,600 million USD. Thus, the corporate name representing a company has considerable influence on its value. A corporate name is the best form of expressing an organization's identity; hence, the corporate name can be a representative means of communicating a company's identity concisely and clearly to internal and external stakeholders [1,2]. Thus, the decision to change the corporate name can have significant strategic effects in areas such as finance and marketing [2]. Changing the corporate name is not only a critical decision but also a result of policy discussions, and its effect is long-lasting [3]. Therefore, thousands of companies worldwide change their names every year to increase their sustainability at a strategic level. Sustainability in this study represents the viability of a company. In particular, IT companies are establishing and implementing management strategies in various ways to survive in a rapidly changing environment. In this respect, a corporate name change is one of the strategies for pursuing sustainability in terms of corporate survival. From this standpoint, the importance of corporate name change is highlighted as one of several factors that affect

company value. Because changing the corporate name can typify a positive signal for its stakeholders, the company can gain economic benefits therefrom [4]. Stakeholders who recognized the announcement of the change of company name as a sign of Good News will try to purchase the company's shares, which will increase the company's stock price. At this time, if a company tries to increase the paid-in capital, it may be advantageous when raising capital because the base price is formed based on the increased stock price. If the stock price, which is a substitute for a company's value, rises, the company's debt raising can also be smooth. Therefore, changing a corporate name can be important for company managers and company stockholders.

In Korea, the number of listed companies that change their corporate names has been increasing annually according to the Korea Securities Depository [5]. In 2021, 122 of the listed companies—39 firms in the Korea Composite Stock Price Index (KOSPI) and 83 firms in the Korea Securities Dealers Automated Quotation (KOSDAQ)—changed their corporate names, representing an increase of 28.4% (27 firms) from the previous year (95 firms). On the one hand, it is noteworthy that the number of companies in the KOSDAQ market (centered on venture companies in Korea's stock market) that have changed their names is more than twice that of the KOSPI market. On the other hand, the number of companies in the KOSDAQ market (centered on venture companies in Korea's stock markets) that have changed their names is more than twice that of the KOSPI market, and 40% to 60% of the annual corporate name changes have been undertaken by companies related to the information technology (IT) industry [6]. As such, many IT companies continue to change their corporate names for their own purposes, but there have been few clear empirical studies behind the incentives of changing their names. Why do IT companies frequently change their corporate name? A company changes its name to improve its image, as well as to change its industry, structure, legal status, and so on [7]. In Korea, when announcing their corporate name change, companies present improvement of company image, business diversification, company partition, company merger, business goals, strategic enhancement, change of largest shareholder, incorporation of affiliation, among others, as their reasons for the change [5]. Moreover, it has been shown that they change their names for one or more reasons [5]. Previous studies on corporate name change have only analyzed the reasons presented by the companies [7–9] and have focused mainly on the stock price response to the name change [6,9–14], and the stock price response to name changes to specific terms (e.g., ".com," "blockchain," "bitcoin," "AI," etc.) [1,15–21]. Meanwhile, merely a few studies have focused on financial performance. These studies have generally analyzed changes in the fragmented financial ratios [6,9,10,22] before and after corporate name changes, but there are few studies that comprehensively analyze the financial improvement effect from a corporate sustainability perspective. The aforementioned gaps in the literature underscore our motivation to study this subject. Although Korea aims to become an IT powerhouse and grows related industries, there are not many studies that analyzed the effectiveness of management strategies on IT companies. However, since Korea opened its capital market in 2000, much attention has been paid to Korean companies. Therefore, in this paper, we comprehensively analyze whether the incentives for the corporate name change of Korean IT companies are to increase corporate sustainability with respect to financial performance (relief of financial constraints). Specifically, this study aims to examine whether there is a significant relationship between the market reactions due to announcements of corporate name changes and the improvement of firms' financial constraints.

This study differentiates itself from previous studies by measuring the effectiveness of a corporate name change from a more comprehensive perspective. In addition, this study is meaningful in that it can specify whether there is a financial improvement effect at a comprehensive level beyond the cosmetic effect of changing the corporate name to those who claim only the weakness of the investor mania hypothesis. Moreover, this study will provide important guidelines to managers of companies considering changing the corporate name and to domestic and overseas investors who want to invest in them by

presenting a significant effect of changing the corporate name as a management strategy for Korean IT companies.

The remainder of this paper is organized as follows: Section 2 reviews previous studies and derives hypotheses, Section 3 discusses the study design and samples, Section 4 presents the results of the empirical analysis, and Section 5 concludes.

## 2. Literature Review and Hypothesis Development

### 2.1. Literature Review

Previous studies on corporate name change can be divided into two main categories: studies on the effect of corporate name change on company value, that is, market response studies according to the rational pricing hypothesis and investor mania hypothesis.

In terms of the rational pricing hypothesis, studies that analyze the effect of the corporate name change on company value can be divided into three categories as follows: first, some studies report that a corporate name change increases corporate value [1,10–12,23,24]. Karim (2004–2007, 38 announcements, event study) [23] examines French companies, Biktimirov et al. (1997–2011, 274~359 firms, event study) [24] investigate Canadian companies and US companies, while Agnihotri and Bhattacharya (2005–2014, 415 firms, event study and OLS regression) [11] and Rani and Asija (2013–2014, 41 firms, event study and logistic regression) [12] focus on Indian companies and find positive market responses to the announcements of corporate name change. Park (2000–2016, 508 firms, event study and OLS regression) [10] and Park et al. (2000–2019, 269 firms, event study and OLS regression) [1] report that the effects of corporate name change are maintained not only for a short period but also for a long period of time for Korean companies. Meanwhile, some studies find that a corporate name change actually decreases company value [3,14,25–27]. Josev et al. (1995–1999, 107 firms, event study and OLS regression) [25] on Australian companies, Mase (1994–2004, 244 firms, event study) [26], Andrikopoulos et al. (1987–2002, 803 events, event study) [3] on British companies, and Devos et al.(1987–2017, 251 firms, event study and OLS regression) [27] on US companies all pay attention to the fact that investors react negatively to a corporate name change. Feng et al. (2007–2017, 760 firms, event study and probit regression) [14] report that the change in the name of Chinese companies has a significantly negative effect on the long-term stock return of companies. However, other studies attest to the lack of relationship between a corporate name change and corporate value [9,28–31]. Howe [28], focusing on US companies, failed to find any significant connection between the announcement of a corporate name change and its stock price from 1962 to 1980. Bosch and Hirschey (1979–1986, 79 firms, event study) [29] and Karpoff and Rankine (1998–1999, 95 firms, event study) [19] examine US companies and report that the corporate name change strategy has no significant effect on the company's stock price. Karbhari et al. (1984–1996, 18 firms, event study) [31], who examine Malaysian companies, have arrived at the same conclusion. Kot (1999–2008, 529 firms, event study and OLS regression) [9] finds that, in the long run, corporate name change and stock prices are uncorrelated.

Changing the corporate name is generally interpreted as the result of the management's strategic decision toward increasing company value. Company managers can sometimes change their corporate names due to specific intentions. Many analysts argue that investors have preferences for certain types of corporate names and that corporate names influence stock prices [31]. This can be explained through the investor mania hypothesis which states that investors want companies to enter a hot market at any cost, and thus, a positive market response to the name change is important in the process of entering a hot market [19]. An example of this hypothesis can be seen around the year 2000, when many companies in the US NASDAQ market changed their corporate names to include the term ".com." Studies on corporate name change based on the investor mania hypothesis were conducted against this backdrop and can be summarized as follows: On the one hand, several studies analyze the market response to this, through which they find that around the day of the announcement of the corporate name change, the stock prices increase significantly [19,20]. On the other hand, a follow-up study conducted at the time of the Internet bubble burst finds that the

changing of corporate names by IT companies positively affects stock prices, but only for those who change their names to exclude the terms ".com," ".net," or "internet" [21]. Moreover, there is some support for this position. Recently, studies have been conducted to analyze the change in corporate value for companies that have changed their corporate names in relation to FinTech, blockchain, cryptocurrency, and artificial intelligence (AI) [13,15–18,32,33]. These studies have found that such changes have had a significantly positive stock price response [13,15–18,32,33]. Jain and Jain [15] found that companies that changed their names to include the words "blockchain" or "bitcoin" saw significant positive abnormal returns that lasted two months after the announcement of the corporate name change. Studies by Cahill et al. [33], Akyildirim et al. [16], Sharma et. al. [17], and Chen et al. [13] reach the same conclusion. Other studies find that companies that are not related to cryptocurrency also have significant positive abnormal returns when they add such words to their corporate name change announcements [32]. Similar conclusions are drawn during the 2000s' ".com" trend [19,20]. Wu and Chen [18] find that constituent stocks of AI exchange-traded funds (ETFs) that have the name "AI" see tremendous positive abnormal returns, implying that the inclusion of certain words in ETF names could create favorable premiums from the name for constituent stocks.

As the abovementioned studies have investigated how investors perceive the company's corporate name change, it is necessary also to examine the internal changes of companies due to the corporate name change. Internal changes in a company are measured by the financial status and management performance, and there are some previous studies related to corporate name change [6,9,10,22]. Oh [22] reports that as the performance of companies improves owing to the corporate name change, their sales increase, but at the same time, there is no synergistic effect on operating profit due to a large increase in costs. Kot [9] reports that changing corporate names is unrelated to business performance. Park [10] finds that companies' total assets, liabilities, equity, current liabilities, sales, and market capitalization significantly increase with corporate name change but see no significant effect on business profits. Rather, Park [10] finds that the net profit decreased significantly. Park and Park's study [6] also arrives at the same conclusion. However, these studies only analyze the partial changes in the financial ratio before and after the name change.

Many companies change their corporate names for corporate sustainability, and many researchers have been analyzing the effects of this strategy. However, research that comprehensively analyzes the financial improvement effect in terms of corporate sustainability is limited. Therefore, this study investigates whether the stock market's response to the disclosure of the change of the company name contributes to the financial improvement effect of the company after the change of the company name to establish whether a company's corporate name change strategy has the effect of overall financial improvement.

## 2.2. Hypothesis Development

Owing to the 4th Industrial Revolution, the management uncertainty of IT companies is increasing, and the pace of change is accelerating. Thus, IT companies must constantly strive for survival during this environmental change. In particular, IT companies are likely to face financial constraints due to continuous research and development, and in this case, they may have to give up important investment opportunities. Financial constraints are usually entailed when companies experience increases in costs when raising capital from outside or have difficulty in raising capital from outside [34,35]. Companies that experience financial constraints seek a variety of ways to escape the situation [36]. If these companies are faced with such situations, financial constraints could be alleviated by changing the corporate name [36]. A corporate name is an essential component of organizational improvement and communicates the value of the company to the market [7]. It is an important way of representing the company's image and increasing market demand for the company [37]. Therefore, if a company changes its name, it can not only benefit from raising capital but also increase the sustainability of the company.

A financial constraint is a factor that constricts investment decisions, such as changing the corporate name, and significantly affects investment decisions [34,35]. Most companies with financial restraints have difficulties in securing external capital. Therefore, there is a possibility of delaying or losing good investment opportunities. If the company decides to delay investment, there is also the risk of being unable to implement the investment plan if the necessary internal capital is not secured within a certain period of time. In other words, financial constraints increase the uncertainty of investment and complicate the making of long-term investments, thereby decreasing corporate sustainability. Thus, companies with greater financial constraints are more likely to implement the strategy to change their corporate names to raise funds more easily [36].

If company's financial constraints are high, the likelihood of the company going bankrupt increases [38]. The greater the chances for the company to go bankrupt, the more restrictions there will be in the process of raising capital. Therefore, these companies will choose to enhance their corporate image through the strategic decision to change their corporate names, through which the likelihood of bankruptcy could be lowered. Recent studies on the stock market response to the announcement of the corporate name change have found that changing the corporate name has a positive effect on stock prices [1,6,11,15–17]. This shows that the market perceives a corporate name change as a favorable factor and strategic choice to increase the company's future sustainability. If the stock prices rise due to the corporate name change, its corporate sustainability may improve. Therefore, the corporate name change will improve the corporate image and thus raise the corporate value, consequently facilitating fundraising to ultimately improve the financial constraints of the company. Accordingly, this study establishes the following hypothesis:

**Hypothesis 1 (H1).** *If a positive market response is shown toward an IT company's corporate name change, its financial constraints will improve.*

Notably, IT companies will try to improve on their current financial constraints for their corporate sustainability, and will want to easily raise capital through the implementation of the strategy to change their corporate names. When a corporate name change is announced, investors evaluate not only the information made public and the signals but also the evidence behind the change [39]. A corporate name change involves costs and risks (e.g., customers' reaction to the new corporate name), and thus, the signal of the name change must be justifiable [39,40]. If the shareholders believe that the announcement to change the corporate name is justified, necessary, and important in generating positive cash flow in the future, their responses would be more favorable [32]. Therefore, a company will work endlessly to induce a positive response from the market toward the announcement to change the corporate name. Accordingly, this study believes that raising capital will be facilitated and the company's financial constraint will be improved after its corporate name is changed if the market response is positive after the announcement for the same, even if the financial constraint before the name change has been dire. Therefore, the following research hypothesis is proposed.

**Hypothesis 2 (H2).** *Although the level of the financial constraints of IT companies before the corporate name change differ, the level of financial constraints will improve after a name change depending on the market response to the announcement of the corporate name change.*

## 3. Research Design and Samples

### 3.1. Variables

#### 3.1.1. Measures for Financial Constraints

This study applies Altman's [38,41] Z-score to measure financial constraint, which is the dependent variable, and aims to analyze the effect of the market response to the strategy to change a corporate name on the financial constraint after the change [42,43]. Furthermore,

we analyze the effect of market response to the corporate name change on the financial constraint by measuring the Z-score of the year before the corporate name is changed, divided into three groups as follows: Group 1 (LowZ$_{FC}$Dummy), with a low financial constraint, is a company with a Z score of 2.99 or higher; Group 2 (MiddleZ$_{FC}$Dummy), with a moderate financial constraint, is a company with a Z score between 1.81 and 2.99; and Group 3 (HighZ$_{FC}$Dummy), with a high financial constraint, is a company with a Z score of 1.8 or lower.

### 3.1.2. Measures for Cumulative Abnormal Return as Market Reaction

The main explanatory variable for testing the hypotheses is the cumulative abnormal return (CAR), which indicates the market response to the corporate name change. The effect of the stock market response to the corporate name change can be measured through an analysis using the event study methodology. The day the announcement to change the corporate name is made is set as the event date to estimate the CAR of each company's stocks. Event study methodology is an empirical analysis method to determine whether the value calculated by estimating the abnormal rate around the time when a specific event is made public to the stock market is due to an announced event (information).

We implement the event study analysis, following Park and Park [6] and Park [10]. This is one of the many empirical research methods that are mostly used in the field of accounting and finance ever since Fama et al.'s [44] study, and it is an empirically verified methodology that is used to determine the amount of change in corporate value due to a specific event [45]. This study uses the market-adjusted return model (MARM) [1,6,10,19,21] to initially calculate each company's abnormal return to estimate CAR. We calculate the abnormal returns of an individual company using Equation (1).

$$AR_{i,t} = R_{i,t} - R_{m,t,} \tag{1}$$

where $AR_{i,t}$ is the abnormal return on day $t$ of the individual company $i$, $R_{i,t}$ is the return on day $t$ of the individual company $i$, and $R_{m,t}$ is the KOSDAQ index return on day $t$. The reason why we use the MARM, which is used by Park et al. [1], Park and Park [6], Cooper et al. [19], and Cooper et al. [21], is to minimize errors caused by using the market model (MM). This is because the result using the MM differs depending on how the estimation period and event period are set [1,6]; that is, in the case of the MM, the CAR of the event period may vary depending on the estimated period setting.

$CAR_{\pm 240}$ is calculated via Equation (2) using the $AR_{i,t}$ of each company calculated with Equation (1). The CAR during ±240 days around the event date is measured [1,6,10]. When estimating long-term returns, eliminating confounding events is unnecessary [46]. The same method is applied to $CAR_{\pm 240}$, the explanatory variable of all the models in this study.

$$CAR_{i,\pm 240} = \sum_{t=-240}^{t=+240} AR_{i,t,} \tag{2}$$

where $CAR_{i,\pm 240}$ is the CAR during ±240 days (a total of 481 days) of the individual company $i$ around the event date, and $AR_{i,t}$ is the abnormal return on $t$ days of the individual company $i$. The reason why we use the event period of ±240 days in this study is that previous studies [1,6,10] have used it relatively recently to analyze the long-term effects of corporate name changes. Figure 1 shows the CAR of our entire sample during ±240 days and also shows the CARs by groups. Referring to Figure 1 below, it can be seen that this period setting well reflects the semi-strong efficient market as an effect on the announcement of the corporate name change. In other words, all the effects are reflected in the market before the announcement date of the change of the company name, and it appears that the effects have not disappeared and have been maintained since the announcement date.

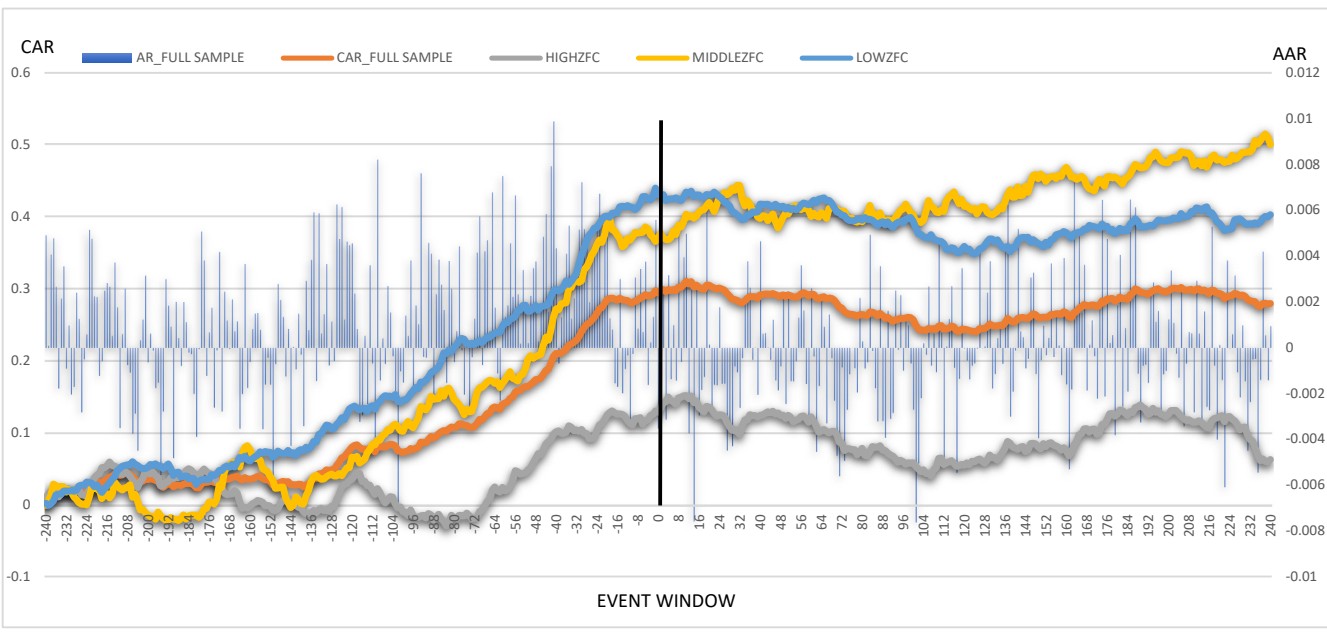

**Figure 1.** CARs during the event date before 240 to after 240.

### 3.1.3. Control Variables

This study controls and analyzes the related variables that can affect the substitute value, $Z_{FC}$, for financial constraints, the dependent variable of this study. The control variables can be explained as follows [1,6,8,10,13,20,27,47,48]: Size is the substitute value for the company's size, and it is obtained by taking the natural logarithm of the sales in the previous year [20,48]. If the company's size is big, information asymmetry can be relatively lower. Therefore, the smaller the company's size, the higher the financial constraint. LiquidityRatio is a substitute value for the company's short-term stability, and it is gained by dividing current assets by current liabilities [1,6,10]. It is an index that measures the capacity of cash mobilization, which is indicative of the company's short-term solvency, and it is a ratio that measures the stability of the financial structure. If a company does not have the short-term financial resources to pay off its debt, it will have difficulty paying off its long-term debts. Therefore, the lower the LiquidityRatio, the worse the financial constraints. Leverage refers to the debt ratio as the substitute value for the company's risk, calculated by dividing total liabilities by total assets as of the end of the previous year [1,6,8,10,13]. The higher the debt ratio, the higher the risk, leading to greater financial constraints from the corporate name change. Furthermore, ROA is the substitute value for the company's profitability, and it is calculated by dividing the business profit by total assets. The lower the business performance (ROA), the worse the financial constraint due to the corporate name change. CEOPrediction is the substitute value for the internal information of a company that is about to change its corporate name that is used to control the effects of the management's psychology of expectation about the company's business profits. It is calculated by subtracting the business profits of the year prior to the corporate name change from the business profits of the year when the corporate name is changed, and dividing that value by the total assets of the year when the corporate name is changed [1,6,10]. If the management's expectations vis-à-vis business performance from the corporate name change are high, it is predicted that the financial constraints will improve after the name change. SalesGrowth is the rate of growth in sales, and the lower the management performance, the higher the probability that the financial constraints will worsen due to the corporate name change [8]. Foreign is the foreigners' share ratio. It is found that foreign shareholders monitor and keep the management's acts in check [47]. Therefore, the higher this value, the lower the financial constraints are predicted to be. LargestSholder is the substitute value for major shareholders' share ratio, and it is the sum

of the holdings of shareholders with more than 5% shares. The higher this value, the more the financial constraint will improve because it will monitor the management. Lossdummy is a dummy variable, as the substitute value of a company's loss. If Lossdummy is 1 [27], this means the financial constraint of a company can worsen after the corporate name change if the company has incurred a loss. Lastly, the level of the financial constraint from changing the corporate name may differ depending on the timing of the corporate name change and the affiliated industry; thus, the year (year) and industry type (industry) have been included in the research model as dummy variables. The definitions of the variables involved in the empirical test in this study are presented in Table 1.

**Table 1.** Variable description.

| Variable Name | Definition and Measurement |
|---|---|
| $Z_{FC}$ | Financial constraints (FC); measured Altman's (1968) Z-Score; as measured by: [1.2 (Working Capital/Total Assets) + 1.4 (Retained Earnings/Total Assets) + 3.3 (Earnings Before Interest and Tax/Total Assets) + 0.6 (Market Value of Equity/Book Value of Total Liabilities) + 1.0 (Sales/Total Assets)]; by the end of the previous year on the announcement date. |
| $CAR_{\pm240}$ | Market Reaction; as measured by: the Cumulative Abnormal Return for event windows from $t-240$ to $t+240$ |
| $LowZ_{FC}Dummy \times CAR_{\pm240}$ | Low Level of Financial Constraints, defined as the interaction term between $LowZ_{FC}Dummy$ and $CAR_{\pm240}$. $LowZ_{FC}Dummy$; a dummy variable; if the $Z_{FCt-1}$ value is greater than or equal to 2.99 then 1, otherwise 0. |
| $MiddleZ_{FC}Dummy \times CAR_{\pm240}$ | Middle Level of Financial Constraints, defined as the interaction term between $MiddleZ_{FC}Dummy$ and $CAR_{\pm240}$. $MiddleZ_{FC}Dummy$; a dummy variable; if the $Z_{FCt-1}$ value is less than 2.99 or more than 1.81 then 1, otherwise 0. |
| $HighZ_{FC}Dummy \times CAR_{\pm240}$ | Hight Level of Financial Constraints, defined as the interaction term between $HighZ_{FC}Dummy$ and $CAR_{\pm240}$. $HighZ_{FC}Dummy$; a dummy variable; if the $Z_{FCt-1}$ value is less 1.81 then 1, otherwise 0. |
| $Size_{t-1}$ | Company's size; as measured by: the natural logarithm of Total Assets$_{t-1}$ |
| $LiquidityRatio_{t-1}$ | Company's Short-Term Stability; as measured by: Current Assets$_{t-1}$/Current Liabilities$_{t-1}$ |
| $Leverage_{t-1}$ | Company's Risk; as measured by: Total Liabilities$_{t-1}$/Total Assets$_{t-1}$ |
| $ROA_{t-1}$ | Company's Profitability; as measured by: Operating Profit$_{t-1}$/ Total Assets$_{t-1}$ |
| $CEOPrediction_t$ | CEO's Prediction; as measured by: $\Delta$ Operating Profit$_t$/Operating Profit$_{t-1}$ |
| $SalesGrowth_{t-1}$ | Year on year growth of net sales; as measured by: $\Delta$Sales$_t$/Sales$_{t-1}$ |
| $Foreign_{t-1}$ | Monitor; as measured by: Foreign Equity$_{t-1}$/Total Equity$_{t-1}$ |
| $LargestSholder_{t-1}$ | Monitor; as measured by: Major Shareholders' Equity$_{t-1}$/Total Equity$_{t-1}$ Major Shareholders having the company's stocks about over 5% |
| LossDummy | Company's loss; a dummy variable that is 1 if loss occurred in the previous year, or 0 otherwise. |
| Year | Year fixed effects |
| Industry | Industry fixed effects |

### 3.2. Research Model Building

This study analyzes whether the corporate name change strategy of IT companies eases financial constraints after the name change depending on the market reactions (H1). This study also analyzes whether the level of financial constraints improves after a name change depending on the market response to the announcement of the corporate name changer, even if the level of the financial constraints of IT companies before the corporate name change differ (H2). Therefore, in this study, the following models, that is, Equations (3) and (4), are established based on Section 3.1 (variables) of this study to verify H1 (Model 1) and H2 (Model 2).

$$[\text{Model 1}]\ Z_{FCt+1} = \beta_0 + \beta_1 CAR_{\pm240} + \beta_{2\sim12}Constrols + \varepsilon \tag{3}$$

$$[\text{Model 2}]\ Z_{FCt+1} = \beta_0 + \beta_1 HighZ_{FC} \times CAR_{\pm240} + \beta_2 MiddleZ_{FC} \times CAR_{\pm240} \\ + \beta_3 MiddleZ_{FC} \times CAR_{\pm240} + \beta_{2\sim12}Controls + \varepsilon \tag{4}$$

### 3.3. Samples

This study analyses the effects of the market responses to the corporate name change strategy on its financial constraint by focusing on the IT companies listed on the KOSDAQ market in Korea from 2010 to 2019. The sample of this study is selected through the following process: we exclude companies that have undergone structural changes, such as

mergers or divisions, and companies that have changed their corporate names two or more times within two years before and after the corporate name change (a total of five years). Furthermore, companies for which the settlement calendar does not end in December and companies for which gathering information is difficult are excluded from the sample.

The final sample, which is collected through this process, comprises 269 cases. The sample data are obtained from the Korea Securities Depository and the Listing Announcement System (http://kind.krx.co.kr, accessed on 5 June 2022) of Korea Exchange. The financial data and the stock price data needed for analysis are extracted from the KIS-Value database of Korea Investors Service.

The sample is composed of 269 cases: 104 cases in the group with a low level (Low$Z_{FC}$) of financial constraints relative to the previous level before the corporate name change, 48 cases in the group with a medium level (Middle$Z_{FC}$) of financial constraints, and 117 cases in the group with a high level (High$Z_{FC}$) of financial constraints. Table 2 presents the distributions of the samples by year.

**Table 2.** Distribution of the samples by year.

| Year | Full Samples | | Level of Financial Constraints | | | | | |
|---|---|---|---|---|---|---|---|---|
| | | | High $Z_{FC}$ | | Middle $Z_{FC}$ | | Low $Z_{FC}$ | |
| | No. | % | No. | % | No. | % | No. | % |
| 2000 | 16 | 5.95 | 1 | 0.96 | 2 | 4.17 | 13 | 11.11 |
| 2001 | 10 | 3.72 | 1 | 0.96 | 4 | 8.33 | 5 | 4.27 |
| 2002 | 11 | 4.09 | 5 | 4.81 | 3 | 6.25 | 3 | 2.56 |
| 2003 | 7 | 2.60 | 3 | 2.88 | 1 | 2.08 | 3 | 2.56 |
| 2004 | 19 | 7.06 | 9 | 8.65 | 4 | 8.33 | 6 | 5.13 |
| 2005 | 22 | 8.18 | 13 | 12.50 | 3 | 6.25 | 6 | 5.13 |
| 2006 | 14 | 5.20 | 4 | 3.85 | 6 | 12.50 | 4 | 3.42 |
| 2007 | 11 | 4.09 | 4 | 3.85 | 2 | 4.17 | 5 | 4.27 |
| 2008 | 10 | 3.72 | 2 | 1.92 | 2 | 4.17 | 6 | 5.13 |
| 2009 | 15 | 5.58 | 9 | 8.65 | 2 | 4.17 | 4 | 3.42 |
| 2010 | 13 | 4.83 | 5 | 4.81 | 4 | 8.33 | 4 | 3.42 |
| 2011 | 11 | 4.09 | 3 | 2.88 | 2 | 4.17 | 6 | 5.13 |
| 2012 | 12 | 4.46 | 5 | 4.81 | 2 | 4.17 | 5 | 4.27 |
| 2013 | 10 | 3.72 | 5 | 4.81 | 1 | 2.08 | 4 | 3.42 |
| 2014 | 8 | 2.97 | 4 | 3.85 | 0 | 0.00 | 4 | 3.42 |
| 2015 | 20 | 7.43 | 11 | 10.58 | 2 | 4.17 | 7 | 5.98 |
| 2016 | 18 | 6.69 | 8 | 7.69 | 2 | 4.17 | 8 | 6.84 |
| 2017 | 15 | 5.58 | 4 | 3.85 | 3 | 6.25 | 8 | 6.84 |
| 2018 | 10 | 3.72 | 2 | 1.92 | 2 | 4.17 | 6 | 5.13 |
| 2019 | 17 | 6.32 | 6 | 5.77 | 1 | 2.08 | 10 | 8.55 |
| Total | 269 | 100 | 104 | 100.00 | 48 | 100.00 | 117 | 100.00 |

## 4. Empirical Findings

### 4.1. Descriptive Statistics

The descriptive statistics of the variables used in this study are presented in Table 3. We winsorize all variables at a 1% level to prevent the contamination of the characteristics of variables by outliers when verifying the hypotheses.

**Table 3.** Descriptive statistics.

| Variable (Obs. = 269) | Mean | t-Value | | SD | 1Q | 2Q | 3Q | Skew | Kurt |
|---|---|---|---|---|---|---|---|---|---|
| $Z_{FCt+1}$ | 2.847 | 11.635 | *** | 4.014 | 0.820 | 2.258 | 3.934 | 1.700 | 4.611 |
| $CAR_{\pm 240}$ | 0.275 | 4.529 | *** | 0.996 | −0.345 | 0.204 | 0.915 | 0.084 | 0.368 |
| $Size_{t-1}$ | 6.203 | 103.858 | *** | 0.979 | 5.550 | 6.112 | 6.814 | 0.450 | −0.005 |
| $LiquidityRatio_{t-1}$ | 2.716 | 14.717 | *** | 3.027 | 1.003 | 1.780 | 2.919 | 2.509 | 6.203 |
| $Leverage_{t-1}$ | 0.422 | 31.005 | *** | 0.223 | 0.247 | 0.420 | 0.560 | 0.352 | −0.465 |
| $ROA_{t-1}$ | −0.038 | −3.946 | *** | 0.157 | −0.103 | 0.005 | 0.061 | −1.451 | 2.297 |
| $CEOPrediction$ | 0.022 | 2.352 | ** | 0.152 | −0.047 | 0.004 | 0.066 | 1.161 | 2.993 |
| $SalesGrowth_{t-1}$ | 0.094 | 2.610 | ** | 0.588 | −0.272 | 0.019 | 0.248 | 1.960 | 5.066 |
| $Foreign_{t-1}$ | 3.377 | 9.815 | *** | 5.643 | 0.107 | 0.779 | 3.612 | 2.115 | 3.542 |
| $LargestSholder_{t-1}$ | 31.029 | 31.634 | *** | 16.088 | 18.205 | 30.320 | 43.050 | 0.395 | −0.646 |
| $LossDummy$ | 0.517 | 16.928 | *** | 0.501 | 0.000 | 1.000 | 1.000 | −0.067 | −2.010 |

This table reports the summary statistics (mean denoted as Mean; standard deviation denoted as SD; skewness value denoted as Skew; kurtosis value denoted as Kurt; 1st, 2nd, and 3rd quartiles denoted as Q1, Q2, and Q3, respectively) of our main variable. The variables are followed by the t-value ($H_0$: μ = 0) based on the results of the one-sample test. **, and *** denote the significance at the 10%, 5%, 1%, and 0.1% levels, respectively. Variables that are not defined herein are described in Section 3.1 (Variables) and Table 1 (Variable description). Our observations range from 2000 to 2019. All variables are taken from KIS-Value database of Korea Credit Ratings Co., Ltd. and the listing disclosure system (http://kind.krx.co.kr, accessed on 5 June 2022), and they are winsorized at the 1% and 99% levels.

The correlation coefficients between the variables used in this study are presented in Table 4, which shows that the overall correlation coefficients are low. Furthermore, an analysis of the variance inflation factors (VIFs) shows that the possibility of multicollinearity problems occurring during the regression analysis is low because the VIFs of all variables are less than the cut-off threshold.

**Table 4.** Correlation matrix.

| Variable | (1) | (2) | (3) | (4) | (5) | (6) | (7) | (8) | (9) | (10) |
|---|---|---|---|---|---|---|---|---|---|---|
| (1) $Z_{FCt+1}$ | 1.000 | | | | | | | | | |
| (2) $CAR_{\pm 240}$ | 0.241 * | 1.000 | | | | | | | | |
| (3) $Size_{t-1}$ | 0.145 * | −0.112 | 1.000 | | | | | | | |
| (4) $LiquidityRatio_{t-1}$ | 0.319 * | 0.040 | −0.098 | 1.000 | | | | | | |
| (5) $Leverage_{t-1}$ | −0.345 * | −0.068 | −0.092 | −0.575 * | 1.000 | | | | | |
| (6) $ROA_{t-1}$ | 0.210 * | 0.239 * | 0.436 * | 0.051 | −0.318 * | 1.000 | | | | |
| (7) $CEOPrediction_{t+1}$ | 0.034 | −0.031 | −0.148 * | 0.013 | 0.135 * | −0.539 * | 1.000 | | | |
| (8) $SalesGrowth_{t-1}$ | 0.008 | 0.008 | 0.015 | −0.050 | 0.004 | 0.253 * | −0.121 * | 1.000 | | |
| (9) $Foreign_{t-1}$ | 0.121 * | −0.041 | 0.284 * | 0.023 | −0.008 | 0.150 * | −0.090 | 0.036 | 1.000 | |
| (10) $LargestSholder_{t-1}$ | 0.108 | 0.141 * | 0.211 * | 0.008 | −0.160 * | 0.328 * | −0.081 | 0.028 | −0.005 | 1.000 |
| (11) $LossDummy_{t-1}$ | −0.176 * | −0.161 * | −0.297 * | −0.015 | 0.270 * | −0.625 * | 0.220* | −0.209 * | −0.159 * | −0.290 * |

This table presents the Pearson (pairwise) correlation coefficients between the variables used in the main regression models. * denote the significance at the 5% level. The variables herein are described in Section 3.1 (Variables) and Table 1 (Variable description).

### 4.2. Regression Results and Discussion

The results of analyzing the effect of the market response to the company's strategy to change its corporate name on that company's financial constraint are presented in Table 5. The dependent variable of this study is $Z_{FCt+1}$, the substitute value for the financial constraint after the name change. The variables of interest are the substitute values for market responses to the name change announcements, $CAR_{\pm 240}$ (for H1) and $LowZ_{FC}Dummy \times CAR_{\pm 240}$, $MiddleZ_{FC}Dummy \times CAR_{\pm 240}$, and $HighZ_{FC}Dummy \times CAR_{\pm 240}$ (for H2).

**Table 5.** Results of the analysis.

| Variable | Model 1 | | | Model 2 | | | VIF |
|---|---|---|---|---|---|---|---|
| | Dep. Var = $Z_{FC}$ | | | Dep. Var = $Z_{FC}$ | | | |
| | Coeff. | t-Stat | | Coeff. | t-Stat | | |
| Intercept | 2.245 | 0.994 | | 2.312 | 1.017 | | |
| $CAR_{\pm 240}$ | 0.265 | 3.917 | *** | | | | 1.687 |
| $HighZ_{FC}Dummy \times CAR_{\pm 240}$ | | | | 0.189 | 3.194 | *** | 1.288 |
| $MiddleZ_{FC}Dummy \times CAR_{\pm 240}$ | | | | 0.102 | 1.724 | + | 1.294 |
| $LowZ_{FC}Dummy \times CAR_{\pm 240}$ | | | | 0.156 | 2.282 | * | 1.722 |
| $Size_{t-1}$ | 0.034 | 0.441 | | 0.030 | 0.390 | | 2.197–2.223 |
| $LiquidityRatio_{t-1}$ | 0.221 | 3.076 | ** | 0.224 | 3.086 | *** | 1.909–1.932 |
| $Leverage_{t-1}$ | −0.124 | −1.606 | | −0.124 | −1.600 | | 2.195–2.199 |
| $ROA_{t-1}$ | 0.157 | 1.573 | | 0.160 | 1.591 | | 3.698–3.715 |
| $CEOPrediction_t$ | 0.131 | 1.896 | + | 0.133 | 1.905 | + | 1.756–1.773 |
| $SalesGrowth_{t-1}$ | 0.047 | 0.761 | | 0.049 | 0.785 | | 1.414–1.420 |
| $Foreign_{t-1}$ | 0.036 | 0.598 | | 0.035 | 0.576 | | 1.335–1.353 |
| $LargestSholder_{t-1}$ | 0.031 | 0.485 | | 0.033 | 0.510 | | 1.513–1.521 |
| LossDummy | 0.011 | 0.146 | | 0.011 | 0.141 | | 2.238–2.239 |
| Year | | included | | | Included | | |
| Industry | | included | | | included | | |
| F-value | 2.951 | *** | | 2.821 | *** | | |
| $R^2(\triangle R^2)$ | 0.415 | (0.275) | | 0.416 | (0.268) | | |
| Durbin-Watson | 2.160 | | | 2.152 | | | |

This table presents the results of regression analysis, focusing on Model 1 (H1) and Model 2 (H2) of this study. R-square significance tests are based on F-values. The estimated coefficients are followed by t-values. +, *, ** and *** denote the 10%, 5%, 1%, and 0.1% significance levels, respectively, based on two-tailed tests. The variables herein are described in Section 3.1 (Variables) and Table 1 (Variable description). All models in Table 4 have Durbin–Watson statistics close to 2. Therefore, it is unlikely that there is autocorrelation of the error term. The possibility of multicollinearity problems occurring during the regression analysis is low because the VIFs of all the variables are less than the cut-off threshold.

Model 1 (H1) of Table 4 is the result of analyzing the effect on the financial constraint ($Z_{FCt+1}$) of the IT company after corporate name change by the degree of market response ($CAR_{\pm 240}$) to the announcement of the change of corporate name after controlling the control variables that affect the financial constraints ($Z_{FC}$) after the name change. The analysis results show that the coefficient value of $CAR_{\pm 240}$ is 0.242 (t-value = 3.637, $p < 0.001$), which means there is a very significant positive impact with the level of financial constrictions after the corporate name change. In other words, the better the market response to the announcement of a company's strategy to change its corporate name, the more the company's financial constriction will improve. This can be interpreted to mean that corporate sustainability will increase as the company will be evaluated more positively in the market due to its corporate name change. Based on previous studies that show that corporate value increases due to a corporate name change, this study reveals that the more positive the market reaction, the better the financial constraint of the company after the name change. In other words, a company's name change strategy has a very significant effect on improving corporate sustainability. Based on previous studies that show that corporate value increases due to a corporate name change, this study shows that the more positive the market reaction, the better the financial constraint of the company after the name change. In other words, a company's name change strategy has a very significant effect on improving corporate sustainability.

Considering the control variables analyzed together in Model 1, Liquidityratio, which is a proxy for liquidity (coeff. = 0.223, t-value = 3.164, $p < 0.01$), is found to have a highly significant positive (+) relationship with the financial constraint after the name change. Leverage, which is a proxy for risk (coeff. = −0.133, t-value = −1.752, $p < 0.1$), is found to have a significant negative (−) relationship with the financial constraint after the name change. Essentially, the higher the liquidity and the lower the debt ratio, the better the financial constraints after the corporate name change. However, the remaining control variables do not have statistically significant values.

Model 2 (H2) in Table 4 presents the results of analyzing the impact on the degree of the financial constraint ($Z_{FC,t+1}$) after the name change concerning the level ($LowZ_{FC}Dummy$,

MiddleZ$_{FC}$Dummy, and HighZ$_{FC}$Dummy) of financial constraints in the year prior to the name change and the market response (CAR$_{\pm 240}$) after the name change announcement. The analysis results show as follows: the coefficient value is 0.189 (t-value = 3.194, $p < 0.01$) when the financial constraint level before the name change is high (HighZ$_{FC}$Dummy*CAR$_{\pm 240}$), the coefficient value is 0.102 (t-value = 1.724, $p < 0.10$) when the financial constraint level is medium (MiddleZ$_{FC}$Dummy*CAR$_{\pm 240}$), and the coefficient value is 0.156 (t-value = 2.282, $p < 0.01$) when the financial constraint level is low (LowZ$_{FC}$Dummy*CAR$_{\pm 240}$). Namely, all the three groups have a statistically significant positive relationship with the degree of financial constraint improvement after the market reaction and the corporate name change. Accordingly, it can be interpreted that the better the market response to the announcement of the corporate name change in all the three groups, the better the financial constraint level after the corporate name change. In other words, even if a financial constraint exists before the name change, the corporate name change strategy is significant in terms of the company's sustainability.

Considering the control variables included in Model 2, LiquidityRatio, which is a proxy of liquidity (coeff. = 0.224, t-value = 3.086, $p < 0.01$), has a very significant positive (+) relationship with the financial constraint after the name change, and CEOPrediction, which is a proxy of profitability (coeff. = 0.133, t-value = 1.905, $p < 0.1$), has a relatively significant positive with the financial constraint after the name change. Therefore, the higher the liquidity and the higher the management's expectation on profitability, the better the financial constraint after the corporate name change. However, there were no statistically significant values for the remaining control variables.

Prior research generally focuses on the effects of fragmentary financial ratios [6,9,10,22] on corporate value. Oh [22] reports that as the performance of companies improves owing to the corporate name change, their sales increase, but at the same time, there is no synergistic effect on operating profit due to a large increase in costs. Kot [9] reports that changing corporate names is unrelated to business performance. Park [10] finds that companies' total assets, liabilities, equity, current liabilities, sales, and market capitalization significantly increase with corporate name change but see no significant effect on business profits. Rather, Park [10] finds that the net profit decreased significantly. Park and Park's study [6] also arrives at the same conclusion. However, these studies only analyze the partial changes in the financial ratio before and after the name change. No previous study has comprehensively examined whether the name change improves the degree of financial constraints with regard to corporate sustainability. Therefore, as this study presents evidence that there is a significant relationship between the change of corporate name and the financial constraint improvement, it contributes to the understanding of the effectiveness of the strategy to change the corporate name. Furthermore, the results of this study raise awareness of the importance of the corporate name change as a strategy to maintain the sustainability of an IT company in today's rapidly changing environment. Therefore, this can be the first study to verify that an IT company's strategy to change its corporate name can help overcome its financial c comstraint.

## 5. Conclusions

This study analyzes whether the corporate name change strategy of IT companies eases financial constraints after the name change depending on the market reactions (H1). The study analyzes whether the level of financial constraints improves after a name change depending on the market response to the announcement of the corporate name changer, even if the level of the financial constraints of IT companies before the corporate name change differ (H2). Specifically, it seeks to determine whether corporate sustainability increases due to the corporate name change strategy by analyzing a sample of IT companies listed on the KOSDAQ market from 2010 to 2019, through the event study methodology and OLS regression.

The conclusions of this study are as follows: First, the analysis result shows that if the market response to an IT company's corporate name change is good, its financial

constraint improves after the name change. This result satisfies hypothesis 1 (H1) of our study. Second, even if the companies' financial constraint conditions before the change of corporate name differ, their financial constraints improve after the name change if the market response to their announcements to change their corporate names is positive. This result satisfies hypothesis 2 (H2) of our study. In other words, in terms of corporate sustainability, the financial constraint of a company improves depending on how it establishes its strategy to change its corporate name, as well as the level of market response to the announcement to change its corporate name. This implies that an IT company's strategy to change its corporate name can affect corporate sustainability. Consequently, from the perspective of IT companies, this study serves as a guide for stakeholders' decision-making processes and proves that financial constraints can be improved through a corporate name change strategy.

The implications and contributions of this study are as follows: First, this study differentiated itself from previous studies by applying Altman's financial constraints as dependent variables to measure the effectiveness of a corporate name change from a more comprehensive perspective, and this study more specifically proved the effect of IT companies in the KOSDAQ market. Second, this study is meaningful in that it can specify that there is a financial improvement effect at a comprehensive level beyond the cosmetic effects of changing the corporate name to those who argue only the fragility of the investor mania hypothesis. Therefore, the results of this study can be highly suggestive to domestic and overseas investors who want to invest in Korean IT companies. Third, by highlighting the effects of a corporate name change as a management strategy for Korean IT companies, this study will provide important guidelines to managers of companies considering corporate name changes and domestic and overseas investors who want to invest in them.

However, despite such contributions, this study has its limitations. The limitations of this study and future research directions are as follows: First, the size of the sample for this study is not large due to various limitations in the design of the study. Therefore, a larger sample is necessary for future studies. Second, we limited the scope of this study to IT companies in the KOSDAQ market, but in future studies, it would be meaningful to compare them with the same sectors in the domestic and overseas markets. Third, in this study, the research period was not significantly subdivided, but, in future studies, it would be meaningful to analyze the difference in the effects of a corporate name change by subdividing the research period (e.g., prosperity period and economic crisis period). We will continue to study the effects of a corporate name change by supplementing the above points and make efforts to generalize the validity of corporate name change strategies.

**Author Contributions:** Conceptualization, Y.S. and J.P.; formal analysis, Y.S.; methodology, Y.S.; visualization, J.P.; writing—original draft, Y.S. and J.P.; writing—review and editing, Y.S. and J.P. All authors have read and agreed to the published version of the manuscript.

**Funding:** This paper was supported by Wonkwang University in 2022.

**Institutional Review Board Statement:** Not applicable.

**Informed Consent Statement:** Not applicable.

**Data Availability Statement:** All data that can reproduce the results in this study can be requested from the corresponding author.

**Conflicts of Interest:** The authors declare no conflict of interest.

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
