# Peer review of "Corporate Sustainability and Market Response According to the Name Change Strategy: Focusing on Korean IT Industry Firms"

_sustainability, doi:10.3390/su141912168_

Round 1

Author Response

We appreciate the efforts the reviewers have made to provide feedback on our manuscript and appreciate the insightful comments and valuable suggestions on our paper.
We have completed extensive proof-reading of the paper and attached a certificate for it.
We have completed the revision of the paper reflecting the reviewer's opinion, and please refer to the attached response letter file for more information.

Reviewer 2 Report

The article takes up an interesting research problem, which is to determine whether corporate sustainability increases due to the corporate name change strategy. The undertaken research issue concerns a fairly new area, which has not yet been thoroughly identified through empirical research. Therefore, this article is part of the discussion and is an attempt to fill the gap in knowledge. However, in order to increase its substantive quality, the article should be corrected taking into account the following comments:

1.    A total of 269 companies, which implemented the corporate name change strategy in 2010-2019, were examined. However, analyzed companies belonged to only one sector (i.e. IT sector). Moreover, the research covers only one country (Korea). It is, therefore, necessary to clearly indicate what value this research will have for the international reader? Why are presented results important for stakeholders (shareholders, managers, etc.) from countries other than Korea?

2.   In section 3.1.2. titled ‘Measures for Cumulative Abnormal Return as Market Reaction’ there should be explained why the authors decided to use the cumulated abnormal return. CAR is a method usually used for the short term, while in this study the event window is 481 days. Is it really the best research method for the chosen event window (i.e. 481 days)? Motivate your choice and explain why another methods (e.g. BHAR) were not chosen? The use of CAR as the best method for these studies must be explained. Moreover, it should be also explained why the index model was chosen instead of, for example, the zero model or the classic market model.

3.    In Conclusion, the authors indicate that it is necessary to expand the scope of the study and compare multiple countries. But what about cross-sector comparison? It is necessary to add in Conclusion whether the authors intend to conduct such a study and explain whether and why it is worth conducting this kind of research. (Or maybe such studies have already been carried out? If so, what are the conclusions/findings?)

4.    The layout of the articles is correct. However, there is no discussion in the paper. It is proposed to add a discussion to the section 4.2 and rename this section from ‘Regression Results’ to ‘Regression Results and Discussion’. Are there any studies on this issue in other countries? If so, what are their results and conclusions? Are the results of these studies consistent with those of the authors? If so, to what extent? Research of which authors shows different results than those achieved by the authors of this article? What is the difference?

5.   The article lacks a clear explanation as to why these and not other control variables were selected for the model. Moreover, authors should consider whether the description of variables should be placed at the end of chapter 3.1.3. instead of Appendix. It seems that it would be clearer to the reader. 

6.  The study was conducted for the years 2010-2019. It would be good to divide the research period into the years of prosperity and economic crises. Has such a study been already carried out by the authors of this article? If so, what are the results? Did the research results differ in the years of prosperity and crisis? If such research has not yet been conducted, indicate this in Conclusion as limitation. 

7. There was no definition of corporate sustainability in the literature review. How do different authors define this term? How corporate sustainability is understood by the authors of this paper? (In line 82 there is written: 'corporate sustainability (financial constraints)'. Should corporate sustainability be understood as financial constrains? It is not clear, therefore please explain and extend it). There should be at least one paragraph devoted to definition of corporate sustainability. 

8.  In Abstract, it is suggested to add an information about the research methods used in this paper. 

9.  In line 90, it would be better to say ‘into two main categories’ instead of ‘into two big categories’.

10. In lines 36-37 the authors claim: “As such, the corporate name representing the company has considerable influence”. An influence on what? This sentence is not finished.

11.  In Introduction (lines 46-47) the authors say: “Because changing the corporate name is a positive signal for its stakeholders, the company can gain economic benefits by doing so”. Is it always true? This sentence should be reformulated. And next: ‘So, changing a corporate name can be important for company managers’ – please explain why. Is it important only for managers? The authors should extend this thought and clarify it.

12.  In Literature Review (lines 91-114) the authors present very briefly the results of research conducted by other scientists. There is no key information regarding research conducted in other markets, such as: research sample and period, research method, sector (IT or other?), etc. This should be added to make the literature review complete.

13.  In line 180 there is said ‘A Financial constraint’ but there is no need to capitalize it. There should be written 'A financial constraint'.

14.  There is no literature for the paragraph between the lines 180-189. Authors ought to add some references to this part of the paper.

15.  Despite the fact that the authors presented the main findings in Conclusion, they did not refer directly to the hypotheses, i.e. they did not say whether the hypotheses were satisfied or not.

16.  In Table 3 (correlation matrix), it is suggested to use asterisks (not bold) as done in the other tables. Such a solution will allow for the consistency of markings throughout the text of the article.

17.  In sentences like ‘(...) which means there is a very significant positive relationship with the level of financial constrictions after the corporate name change’ (line 381-382) it is better to use the terms 'impact' or 'influence' instead of 'relationship'. ('Relationship' is usually used to describe the correlation between variables, while 'impact/influence' is used to describe the estimation results).

18.  More emphasis should be placed on presenting the research gap. Why is it worth taking up these research issues? Why is this topic important for economic theory and practice? Add it to Introduction.

Author Response

(The authors gave the same response as above.)

Reviewer 3 Report

Dear authors,

The title seems interesting at first glance, but I witnessed many controversies as I read the manuscript. The aim is not clear, and sometimes I feel that you will answer many diverse questions. Then, you shall re-read and re-write the manuscript to make it more coherent. The following comments could help you improve the early draft:

First, the paper needs extensive English language proofreading.

The referencing style is not in line with the journal's guidelines. Why, on page 2, do you enter [27] after mentioning [4]?

Please provide more facts and figures to show the research gap. The theoretical contribution is not discussed clearly.

The hypotheses are too long and somehow incomprehensible. Please revise them.

Please provide references while mentioning critical issues, such as "but there are few studies that comprehensively analyze the financial improvement effect from the corporate sustainability point of view", on page 2.

Please compare the findings with those of the others. Otherwise, your theoretical contribution remains unclear. On pages 11 and 12, you shall compare the findings with similar and distinct results of other studies.

Please explain the limitations, implications and directions for future research. These issues are briefly mentioned on pages 11 and 12, but it is not enough. You need to go to the details, especially regarding the directions for future research.

Best fo luck!

Author Response

(The authors gave the same response as above.)

Round 2

Reviewer 1 Report

I am satisfied with the revision.

Author Response

We appreciate the efforts the reviewers have made to provide feedback on our manuscript and appreciate the insightful comments and valuable suggestions on our paper.
We have completed the revision of the paper reflecting the reviewer's opinion, and please refer to the attached response letter file for more information.
